# Text-Derived Language Identity Incorporation for End-to-End Code-Switching Speech Recognition

**Qinyi Wang**
National University of Singapore
qinyi@u.nus.edu

**Haizhou Li**
National University of Singapore
The Chinese University of Hong Kong, Shenzhen
haizhouli@cuhk.edu.cn

## Abstract

Recognizing code-switching (CS) speech often presents challenges for an automatic speech recognition system (ASR) due to limited linguistic context in short monolingual segments, resulting in language confusion. To mitigate this issue, language identity (LID) is often integrated into the speech recognition system to provide additional linguistic context. However, previous works predominately focus on extracting language identity from speech signals. We introduce a novel approach to learn language identity from pure text data via a dedicated language identity-language model. Besides, we explore two strategies: LID state fusion and language posterior biasing, to integrate the text-derived language identities into the end-to-end ASR system. By incorporating hypothesized language identities, our ASR system gains crucial contextual cues, effectively capturing language transitions and patterns within code-switched utterances. We conduct speech recognition experiments on the SEAME corpus and demonstrate the effectiveness of our proposed methods. Our results reveal significantly improved transcriptions in code-switching scenarios, underscoring the potential of text-derived LID in enhancing code-switching speech recognition.

## 1 Introduction

Automatic speech recognition (ASR) systems have long grappled with the complex task of accurately transcribing multilingual speech, especially when it involves the phenomenon known as code-switching (CS). Code-switching, or code-mixing, refers to the practice of alternating between two or more languages or dialects within a single conversation. Within these intricate language mixtures, the presence of very short monolingual segments further compounds the challenge. The limited contextual information within these segments often leads to confusion in recognizing phonetically similar words from different languages, substantially affecting the overall performance of multilingual ASR systems (Amazouz et al., 2017; Yılmaz et al., 2018; Wang et al., 2019). Traditional approaches to code-switching ASR have resorted to language identification or code-switching detection systems as separate pre-processing or post-processing steps. While these methods have been effective in providing additional language context (Weiner et al., 2012a; Vu et al., 2012; Zhang, 2013), they introduce additional complexity to the ASR pipeline and increase processing time.

The advancement of deep learning has brought forth a remarkable paradigm shift in multilingual and code-switching ASR systems. End-to-end (E2E) approaches have gained substantial attention, harnessing advanced neural network architectures, such as the Transformer (Vaswani et al., 2017), to directly transcribe code-switched speech (Zhou et al., 2020; Dalmia et al., 2021). These E2E CS ASR systems automatically capture both the acoustic and linguistic characteristics of code-switched speech, eliminating the need for explicit acoustic and language modeling. To reduce language confusion in E2E CS ASR systems, researchers have studied methods for integrating language identity (LID) information learned from both paired speech-text data (Shan et al., 2019; Qiu et al., 2020; Zhang et al., 2021) and unpaired speech data (Li et al., 2019; Punjabi et al., 2020; Tseng et al., 2021). However, these prior approaches have predominantly focused on learning language identities from speech data, overlooking the untapped potential of pure text data. Compared to annotated speech data, text data offers a more accessible and readily available resource.

This paper explores the possibility of learning language-switching patterns from pure text data without relying on speech data. Text-derived features, such as part-of-speech tags and syntactic features, have been shown to improve the performance of code-switching language models (Adel

et al., 2013, 2015; Winata et al., 2018). Inspired by these achievements, we propose to infer language identities from text data and integrate these text-derived language identities into the end-to-end CS ASR system. Our objective is twofold: to acquire language-switching syntax and patterns from text data in the form of language identities, and to leverage these language identities to mitigate the language confusion in code-switching speech recognition. We believe the paired in-domain text inherently encompasses a wealth of linguistic information, which can be effectively harnessed to advance code-switching speech recognition.

To acquire code-switching patterns sorely from text data, we propose a novel language modeling scheme, called language identity-language model, to predict the language identity of the next text token based on a combined history of previous text and language identity tokens. The proposed language identity-language model is jointly trained with a Transformer-based ASR model. Furthermore, we explore two strategies for integrating the predicted language identities into an E2E ASR system: LID state fusion and language posterior biasing. In the LID state fusion strategy, we combine token-level language identity hidden states with ASR hidden states using learned weights. On the other hand, the language posterior biasing strategy directly adjusts the ASR posterior probabilities based on the hypothesized language identities and language posteriors. We evaluate the effectiveness of these proposed methods on the SEAME corpus, a Mandarin-English speech dataset. The results underscore the efficacy of our approaches in reducing language confusion during code-switching speech recognition, leading to more accurate transcriptions of code-switched speech.

The remainder of this paper is structured as follows: Section 2 reviews the background of the Transformer-based ASR with parallel speech-text decoder. Section 3 presents related work for leveraging language identification in multilingual and code-switching speech recognition. Section 4 presents the proposed LID-LM for generating language identities from pure text data and LID integration strategies. Section 5 describes the datasets, models, and evaluation metrics used to assess the performance of the proposed method. Section 6 presents the results and analysis of the experiments. Finally, Section 7 summarizes the contributions and outlines directions for future research.

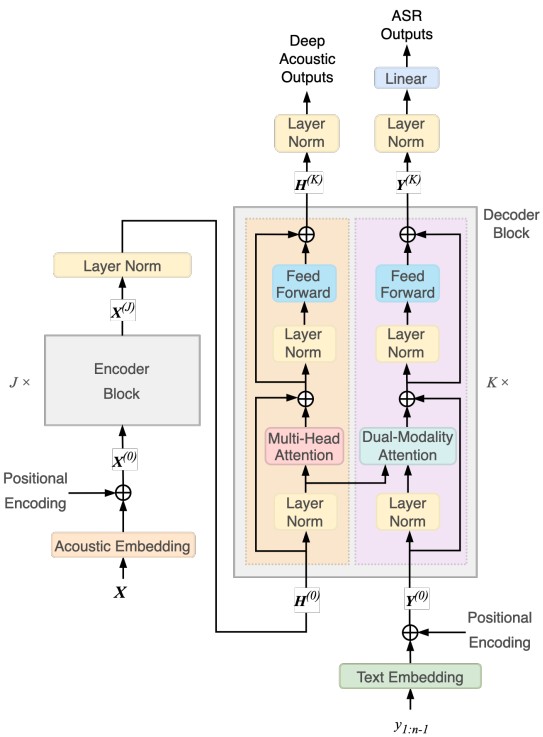

Figure 1: The architecture of a Transformer-based ASR with parallel speech-text decoder.

## 2 Background

In this section, we briefly review the Transformer-based ASR architecture with parallel speech-text decoder that we used to implement our method.

### 2.1 Transformer with Parallel Speech-Text Decoder

We use the Transformer-based ASR with parallel speech-text decoder architecture as our baseline model, as illustrated in Figure 1. This decoder architecture is a variant of the decoder architecture in the speech-and-text Transformer (Wang et al., 2023). Our preliminary studies indicate that this decoder architecture outperforms the vanilla Transformer decoder architecture in both monolingual and code-switching English and Chinese speech recognition tasks.

The parallel speech-text decoder consists of a stack of $K$ identical decoder blocks. Each decoder block comprises two parallel branches: a deep acoustic branch and a speech decoding branch. The inclusion of the deep acoustic branch facilitates the learning of speech-text alignment by projecting the acoustic representations to a comparable level of abstraction as the text representation.

The deep acoustic branch generates new deep

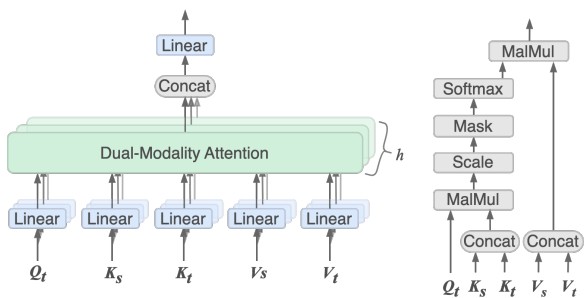

Figure 2: (left) An dual-modality attention module that adopts dual-modality scaled dot-product attention. (right) dual-modality scaled dot-product attention.

acoustic states $\boldsymbol{H}^{(k)}$ by attending to the deep acoustic states from its previous block as follows,

$$\boldsymbol{H}^{(k)} = \text{STDecBlock}(\boldsymbol{H}^{(k-1)}). \quad (1)$$

Meanwhile, the speech decoding branch generates decoder states $\boldsymbol{Y}^{(k)}$ by attending to the deep acoustic states and the decoder states from its previous block as follows,

$$\boldsymbol{Y}^{(k)} = \text{STDecBlock}(\boldsymbol{Y}^{(k-1)}, H^{(k-1)}). \quad (2)$$

Finally, the probability of the next text token, given the complete acoustic feature sequence and its previous text token history, is calculated using the decoder states generated from the last decoder block as follows,

$$P(y_n|\boldsymbol{X}, \boldsymbol{y}_{1:n-1})$$
$$= \text{Softmax}(\text{ Linear }(\text{LayerNorm}(\boldsymbol{Y}^{(K)}))). \quad (3)$$

### 2.2 Dual-Modality Attention

Another key feature of the parallel speech-text decoder is the use of the dual-modality attention mechanism, which is a variation of the on-demand dual-modality attention proposed in our speech-and-text Transformer framework. As depicted in Figure 2, this mechanism establishes dependencies between the text-text representations and text-speech representations through the mapping of a query and two sets of key-value pairs into an output representation. The dual-modality attention is formulated as follows,

$$\text{DualModalityAttention}(\boldsymbol{Q}_t, \boldsymbol{K}_t, \boldsymbol{V}_t, \boldsymbol{K}_s, \boldsymbol{V}_s)$$
$$= \text{Softmax}\left(\frac{Q_t K_c^T}{\sqrt{d}}\right) V_c. \quad (4)$$

This multi-head attention mechanism has five input vectors - a query and two sets of key-value pairs: target query $Q_t$, target key $K_t$, target value $V_t$, source key $K_s$ and source value $V_s$. Here, $K_c$ is the concatenation of $K_t$ and $K_s$, and $V_c$ is the concatenation of $V_t$ and $V_s$.

## 3 Related Work

Language identification plays a vital role in various multilingual speech processing applications, particularly in multilingual or code-switching speech recognition systems. It provides valuable contextual information that regulates speech recognizers and reduces language confusion. Early multilingual ASR systems adopted a two-stage approach, where a language identification component was incorporated at the front-end to distinguish speech from different languages, followed by the use of monolingual recognizers to transcribe speech in specific languages at the back-end (Bhuvanagiri and Kopparapu, 2010; Lyu et al., 2006). Subsequently, frame-level language identities predicted by a dedicated language identification module were integrated into the ASR decoding process to handle rapid language changes (Vu et al., 2012; Weiner et al., 2012b).

With the advancements of deep learning, there has been a shift towards incorporating language identification as an auxiliary task, jointly learned with end-to-end multilingual and code-switching speech recognition systems (Luo et al., 2018; Zeng et al., 2018; Li and Vu, 2019; Yin et al., 2022; Liu et al., 2023). Additionally, Seki et al. (Seki et al., 2018) dynamically track the language identity in code-switching utterances by adding language identity tokens before code-switching points in speech transcriptions, eliminating the need for an external language identification module. This LID token augmentation method is also used in (Zhang et al., 2021), where different language embeddings are concatenated with the word embedding of text tokens to further enhance the distinguishability between word embeddings of different languages.

However, existing research has predominantly concentrated on extracting language identity from speech signals. In contrast, our work takes an innovative approach, delving into the prospect of learning language identity directly from raw text data. By harnessing the rich linguistic information within textual data, we aim to reduce language confusion in end-to-end code-switching speech recognition

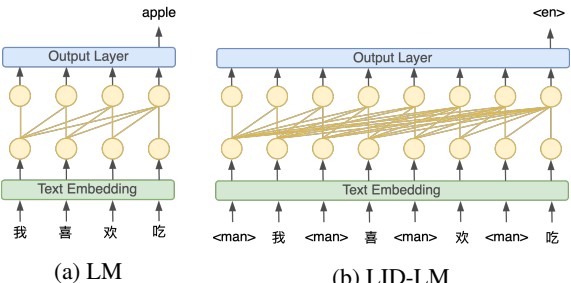

(a) LM        (b) LID-LM

Figure 3: Illustration of LM and LID-LM. LM predicts the next text token given the previous text token history, while LID-LM predicts the next language identity token or the next text token given the previous augmented language identity and text token history.

systems, leading to more accurate transcriptions.

## 4 Proposed Method

### 4.1 Language Identity-Language Model

The goal of language models is to assign probabilities to sequences of words. Let's consider an $N$-length text token sequence represented as as $\boldsymbol{Y} = \{y_1, \ldots, y_n, \ldots, y_N\}$. The aim of a neural language model is to predict the probability distribution $P(y_n|\boldsymbol{y}_{1:n-1})$ over the vocabulary $\mathcal{V}$, given the previous text token histories $y_{1:n-1}$. To explicitly incorporate language identity information into the language model, we introduce a novel language model scheme called language identity language model (LID-LM), which takes the input of token sequences with language identities inserted into the front of each text token. Figure 3 compares the original language model with the proposed LID-LM. The incorporation of language identities into the output token set enriches the language model, enabling it to capture nuanced language-specific characteristics and code-switching patterns from text data. This LID token augmentation technique has been shown to be effective in reducing language confusion in ASR systems (Seki et al., 2018; Zhang et al., 2021). By augmenting language identities into the text token sequences, the LID-LM offers an innovative way to predict language identity without relying on speech data.

To accommodate the language identity information within the LID-LM, we use an augmented vocabulary $\mathcal{V}' = \mathcal{V} \cup \mathcal{V}^{lid}$. Here, $\mathcal{V}^{lid}$ represents the set of language identity tokens. The augmented $2N$-length sequence $\boldsymbol{Z} = \{z_1, \ldots, z_n, \ldots, z_{2N}\}$ corresponds to the alternating arrangement of language identity tokens and text tokens. In this sequence, the odd-indexed tokens represent the language identity tokens, while the even-indexed tokens represent the text tokens. The LID-LM aims to predict the probability of the next text token $z_{2n}$ or the next language identity token $z_{2n-1}$ based on the history sequence of language identity and text tokens $\boldsymbol{z}_{1:2n-1}$ or $\boldsymbol{z}_{1:2n-2}$, respectively.

For optimization, we adopt the cross-entropy loss as the loss function for the LID-LM. This loss function measures the discrepancy between the predicted token distribution and the true label distribution, and it is normalized by the total number of tokens in the training data. We denote this loss as $L_{lid-lm}$. In the subsequent subsections where we describe the LID integration methods, the LID-LM is jointly trained with the ASR model by using the following formulation,

$$\mathcal{L}_{joint} = \alpha\mathcal{L}_{ctc} + (1-\alpha)\mathcal{L}_{att} + \beta\mathcal{L}_{lid-lm}, \quad (5)$$

where $\mathcal{L}_{ctc}$ and $\mathcal{L}_{att}$ represents the CTC loss and label smooth loss for the hybrid CTC/Attention ASR model, and $\mathcal{L}_{lid-lm}$ denotes the cross-entropy loss for the LID-LM. The weights $\alpha$ and $\beta$ control the contribution of each loss component, enabling a balanced optimization for the ASR and LID-LM components.

During the training process, we utilize speech transcriptions augmented with the groundtruth language identities as the input text to train the language identity-language models. This ensures that the LID-LMs learn to associate the correct language identities with the corresponding text tokens. During decoding, we incorporate the previously decoded text token from the ASR model and its corresponding language identity as the previous token history for the LID-LM. This restricts the LID-LM to generate language identity predictions based on the ASR model's previous output transcriptions. In addition, we share the weights of the text embeddings between the ASR and LID-LM models. This parameter sharing enables the ASR model to understand and utilize the language identities provided by the LID-LM during the integration strategies described in the subsequent subsections.

### 4.2 LID State Fusion

The effectiveness of language-specific gating mechanisms in multilingual speech recognition has been demonstrated in a previous work (Kim and Seltzer, 2018). In their approach, the ASR system utilizes

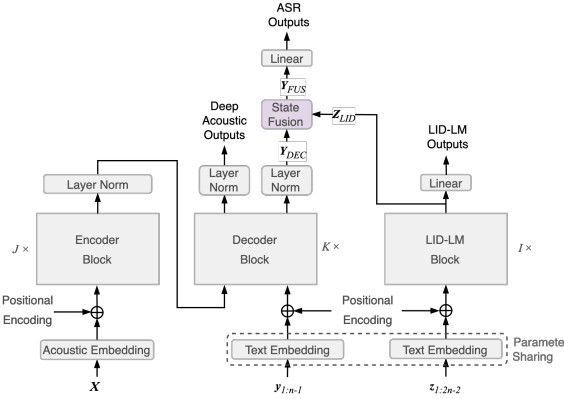

Figure 4: A schematic representation of the LID state fusion method.

one-hot vectors as language indicators to modulate its hidden state in each layer. Building upon this motivation, we propose a novel LID state fusion approach to use a gating mechanism to guide the ASR with language-specific information generated by the LID-LM. Specifically, the ASR system incorporates the language identity information by fusing the language identity hidden states into its own decoder output states. By leveraging the language-specific information provided by the LID-LM, the ASR system can effectively adapt its predictions and enhance its performance in code-switching scenarios according to the linguistic knowledge and confidence level implicitly contained in the language identity hidden states.

The LID state fusion method is illustrated in Figure 4. We generate token-level LID hidden states $Z_{LID}$ from the LID-LM, utilizing the historical token sequence of language identity and text tokens $z_{1:2n-2}$. Simultaneously, we apply layer normalization to the hidden states $\boldsymbol{Y}^{(K)}$ of the speech-text decoder from the last decoder block, generating the normalized decoder hidden states as follows,

$$\boldsymbol{Y}_{DEC} = \text{LayerNorm}(\boldsymbol{Y}^{(K)}). \quad (6)$$

Next, we fuse the token-level LID representation with the normalized decoder's hidden states using a state fusion gate, producing a combined representation $Y_{FUS}$. The state fusion gate employs a gating mechanism inspired by the work of Sriram et al. (Sriram et al., 2018), calculated as follows,

$$\boldsymbol{G} = \text{Sigmoid}(\text{Linear}(\text{Concat}(\boldsymbol{Y}_{DEC}, \boldsymbol{Z}_{LID}))), \quad (7)$$

$$\boldsymbol{Z}_{GATED} = \text{MatMul}(\boldsymbol{G}, \boldsymbol{Z}_{LID}), \quad (8)$$

Table 1: Dataset statistics of the SEAME corpus used in the CS experiments.

| | #Hours | #Utterances | | | |
| | | Mandarin | English | CS | Total |
|---|---|---|---|---|---|
| Train | 96 | 20,313 | 20,283 | 48,342 | 88,938 |
| Dev. | 5 | 1,163 | 1,152 | 2,685 | 5,000 |
| $Eval_{man}$ | 7 | 1,420 | 808 | 4,303 | 6,531 |
| $Eval_{sge}$ | 4 | 500 | 2,656 | 2,165 | 5,321 |

$$\boldsymbol{Y}_{FUS} = \text{Linear}(\text{Concat}(\boldsymbol{Y}_{DEC}, \boldsymbol{Z}_{GATED})). \quad (9)$$

Finally, the output probability of the next text token, given the complete acoustic feature sequence, its previous text token history, and its previous text and LID token history, is calculated as follows,

$$P(y_n|\boldsymbol{X}, \boldsymbol{y}_{1:n-1}, \boldsymbol{z}_{1:2n-1})$$
$$= \text{Softmax}(\text{Linear}(\boldsymbol{Y}_{FUS})). \quad (10)$$

### 4.3 Language Posterior Biasing

The adjustment of the ASR system's posterior probabilities using language posteriors, generated either by external or internal language identification components, has been demonstrated to enhance the performance of code-switching ASR systems (Liu et al., 2023; Tseng et al., 2021; Li et al., 2019). In this study, we investigate the effectiveness of the language posterior biasing method by leveraging token-level language posteriors obtained from the language identity-language model.

In the language posterior biasing method, we first generate the token-level language posterior $P(z_{2n-1}|z z_{1:2n-2})$ using the LID-LM. Then, we adjust the ASR's posterior probabilities by its corresponding language posteriors based on the following formulation,

$$P^{bias}(y_n|\boldsymbol{X}, \boldsymbol{y}_{1:n-1})$$
$$= P(y_n|\boldsymbol{X}, \boldsymbol{y}_{1:n-1}) \times P(z_{2n-1}|\boldsymbol{z}_{1:2n-2}), \quad (11)$$

Here, $P(y_n|\boldsymbol{X}, \boldsymbol{y}_{1:n-1})$ represents the original ASR posterior probability for text token $y_n$. The term $P(z_{2n-1}|\boldsymbol{z}_{1:2n-2})$ corresponds to the language posterior probability for the language identity token $z_{2n-1}$, obtained from the language identity-language model.

## 5 Experimental Setup

### 5.1 Dataset

To assess the effectiveness of the proposed methods, we conduct language identification and code-switching speech recognition experiments on the

SEAME Mandarin-English speech corpus (Lyu et al., 2010). This corpus comprises approximately 110 hours of spontaneous code-switching speech collected from Singapore and Malaysia college students and staff members. The recordings were captured with close-talk microphones during interview and conversation settings. The corpus includes a mix of inter-sentential and intra-sentential code-mixing utterances, as well as monolingual utterances.

In the ASR experiments, we partitioned the SEAME training set into a development set *dev.* and a training set *train*. Specifically, we randomly selected 5,000 utterances to form the *dev.* set, while the remaining paired data was used for the *train* set. It is important to note that the evaluation sets, denoted as $eval_{man}$ and $eval_{sge}$, exhibit varying distributions of monolingual Mandarin, monolingual English, and Mandarin-English code-switching utterances, as summarized in Table 1. The $eval_{man}$ set is primarily composed of monolingual Mandarin and code-switching utterances, while the $eval_{sge}$ set contains a higher proportion of monolingual English utterances.

For the LID experiment, we manually augmented the transcriptions of the above sets by inserting the corresponding language identity token at the beginning of each text token. Given that we are using the Mandarin-English language pair, the distinctions between Chinese characters and the English alphabet are readily discernible. To ensure consistency and standardization, we utilized the default SEAME recipe provided by the end-to-end speech processing toolkit ESPnet (Watanabe et al., 2018) to distinguish between Chinese and English tokens. Consequently, there was no need for inter-annotator agreement within our experimental setup.

## 5.2 Implementation Details

**Dictionary.** To effectively model both English and Chinese languages, as well as their corresponding language identities, we construct a bilingual dictionary that includes additional language identity tokens. For English, we apply the byte-pair encoding (BPE) to the English-only transcriptions of the SEAME *train* set to generate subword units, resulting in a vocabulary size of 3,000 as the modeling units. As for Chinese, we select 5,103 frequently used Chinese characters extracted from the three unpaired text datasets. To represent the language

identities of the text tokens, we introduce LID tokens into the output units. Specifically, we include $\langle en \rangle$, $\langle man \rangle$, and $\langle na \rangle$ tokens to represent the English, Chinese, and other identities, respectively. These LID tokens allow explicitly incorporating language identity information during the modeling process. Additionally, we incorporate special tokens $\langle unk \rangle$, $\langle sos \rangle$, and $\langle eos \rangle$ to handle unknown words, the start of a sentence, and the end of a sentence, respectively.

**Models.** The Transformer-based ASR models used in our experiments consist of 8 encoder layers and 6 parallel speech-text decoder layers. The LID-LM and external LM used for shallow fusion are all 6-layer Transformer decoder models. All models have output dimension of 256, inner-layer dimension of 2,048, and 4 attention heads. For the two proposed LID integration methods, the LID-LM is co-trained with the ASR models from scratch. We employ the Adam optimization algorithm with an initial learning rate of 1.0 for ASR models and $1.0 \times 10^{-4}$ for language models. To schedule the learning rate, we use the Noam learning rate scheduler (Vaswani et al., 2017) with 25,000 warmup steps for ASR models and use the cosine decay scheduler (Loshchilov and Hutter, 2016) for LMs with 1,000 initial steps and 100,000 total steps. Dropout regularization with a rate of 0.1 is applied to all models to prevent overfitting. All models are trained for 50 epochs. The CTC weight $\alpha$ is set to 0.3 for all models during training and set to 0.5 during decoding. The LID-LM weight $\beta$ is set to 0.7 during training. To select the best model for inference, we average the parameters from the top 10 epochs based on their performance on the validation set.

**Evaluation.** For LID-LM, we report its token-level language identification accuracy. For ASR evaluation, we use the Mix Error Rate (MER) as the performance metric. MER combines the Word Error Rate (WER) for English tokens and the Character Error Rate (CER) for Chinese tokens. This evaluation metric provides a comprehensive assessment of the ASR models' accuracy in transcribing both English and Chinese languages within the Mandarin-English code-switching context.

## 6 Results and Discussion

### 6.1 Language Identification Accuracy

We begin by assessing the language identification performance of the LID-LM. The LID-LM, trained

Table 2: SEAME: MERs on the $eval_{man}$ and $eval_{sge}$ set. Upper section: E2E ASR systems with LID integration methods discussed in Section IV. Lower section: Transformer-based ASR model with our proposed LID integration methods.

| Model | Method | MER% | |
| --- | --- | --- | --- |
| | | $Eval_{man}$ | $Eval_{sge}$ |
| GRU-based Encoder-Decoder (Luo et al., 2018) | Baseline | 35.4 | 37.8 |
| | LID Joint Learning | 34.1 | 36.5 |
| BLSTM-based Encoder-Decoder (Zeng et al., 2018) | Baseline | 26.4 | 36.1 |
| | LID Joint Learning | 26.0 | 35.8 |
| LSTM-based Transducer (Zhang et al., 2021) | Baseline | 33.3 | 44.9 |
| | CS Point Tagged Text | 30.2 | 41.5 |
| Transformer-based Encoder-Decoder with Parallel Speech-Text Decoder | Baseline | 21.4 | 29.5 |
| | Shallow Fusion | 21.0 | 29.0 |
| | LID State Fusion | **20.4** | **28.2** |
| | Language Posterior Biasing | 21.3 | 29.2 |

using the augmented transcriptions of the SEAME training set, achieves token-level LID accuracies of 78.7% and 80.1% on the $eval_{man}$ and $eval_{sge}$ sets, respectively. These accuracies surpass the previously reported LID accuracy of 70.6% in (Weiner et al., 2012b) that relies on acoustic features to generate frame-level LID predictions. This outcome underscores the richness of linguistic information present in text-only data, indicating its capacity to provide valuable linguistic clues pertaining to grammar and language switching patterns. Thus, our findings emphasize the potential of leveraging LID-LM for accurate and robust language identification in code-switching speech recognition systems.

### 6.2 ASR Results

We proceed to evaluate the speech recognition performance of various methods on the SEAME evaluation sets, namely $eval_{man}$ and $eval_{sge}$. The upper section of Table 2 presents the performances of previous LID integration methods for end-to-end ASR systems. It can be observed that these methods result in minor to moderate improvements on the SEAME evaluation sets. Among them, the CS point tagged text method proposed by Zhang et al. (Zhang et al., 2021) achieves the most significant improvement, with relative MER reductions of 9.3% and 7.6% on the $eval_{man}$ and $eval_{sge}$ sets, respectively.

The lower section of Table 2 summarizes the performances of our Transformer-based ASR system and our proposed LID integration methods. The Transformer-based ASR baseline achieves MERs of 21.4% and 29.5% on the $eval_{man}$ and $eval_{sge}$ sets, respectively. Shallow fusion with an external language model trained on the training transcrip-

tions leads to slight improvements of 1.9% and 1.7% over the baseline ASR system. Our proposed LID state fusion method demonstrates moderate relative MER reductions of 4.7% and 4.4% on the $eval_{man}$ and $eval_{sge}$ sets, respectively. However, the language posterior biasing method only yields marginal improvements to the baseline ASR system.

The above results suggest that the learned gating parameter in LID state fusion plays a crucial role in enabling the ASR system to effectively incorporate and utilize the contextual cues provided by the language identity-language model. By dynamically adjusting the contribution of the LID hidden states, the ASR system is better at capturing language-specific patterns and transitions, resulting in improved code-switching speech recognition performance. In contrast, the LID posterior fusion method lacks such an automatic adjusting mechanism, making the system vulnerable to error propagation in the language identification module. As a result, the effectiveness of leveraging language identification to enhance speech recognition performance is hindered in this approach.

### 6.3 Error Analysis

To gain insights into the speech recognition performances of the Transformer-based ASR model and the ASR model with the LID state fusion method, we conducted an analysis of the recognition errors across three different utterance categories: monolingual Mandarin, monolingual English, and code-switching Mandarin-English. Figure 5 presents the error counts for these categories on the $dev_{man}$ and $dev_{sge}$ sets.

The figure shows that the employment of the LID

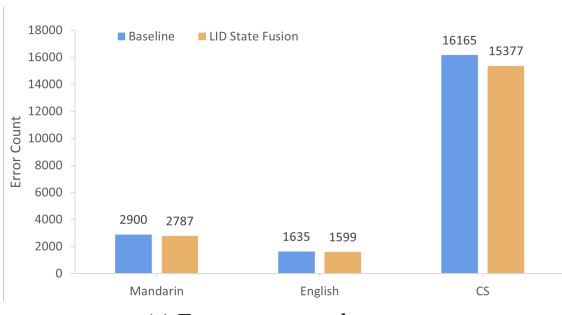

(a) Error counts on $dev_{man}$ set

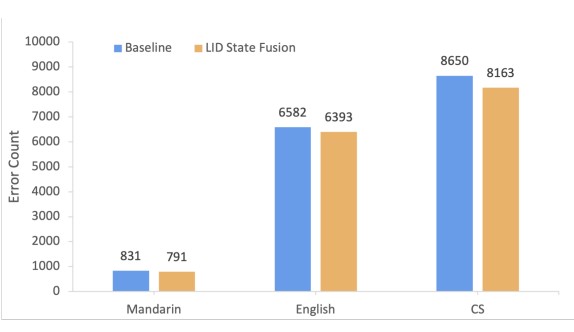

(b) Error counts on $dev_{sge}$ set

Figure 5: Error counts on $dev_{man}$ and $dev_{sge}$ sets across three utterance categories. For Mandarin utterances, the error counts refer to character error counts. For English utterances, the error counts refer to word error counts. For code-switching utterances, the error counts refer to mixed error counts.

Table 3: Examples of references and transcriptions generated by the baseline and fusion models. Recognizing errors (**** means deletion) are indicated in red color.

| | Transcription |
|---|---|
| Reference | 大概啊那个 cheese 大概二十五这样咯 |
| Baseline | 大概 er 那个 去 大概二十五这样咯 |
| Fusion | 大概啊那个 cheese 大概二十五这样咯 |
| Reference | 圣诞节快到了 have you made any preparations |
| Baseline | 圣诞节快到了 **** every may any preparations |
| Fusion | 圣诞节快到了 have you may any preparations |
| Reference | 我当然是买那个比较好的 right |
| Baseline | 我 大家 是买那个比较好的 right |
| Fusion | 我当然是买那个比较好的 right |
| Reference | twenty percent 还是 two percent 一千不见两百是 |
| Baseline | then 第 percent 还是 two percent |
| Fusion | twenty percent 还是 two percent 一千不见两百是 |
| Reference | 你觉得你很 fit 吗 |
| Baseline | 你觉得你很 fat 吗 |
| Fusion | 你觉得你很 fit 吗 |

state fusion method results in a significant reduction in error counts for code-switching utterances. Specifically, we observed reductions of 4.9% and 5.6% in error counts on the $dev_{man}$ and $dev_{sge}$ sets, respectively. This indicates that the LID state fusion method effectively mitigates language confusion and improves the overall system performance, particularly in code-switching scenarios. Moreover, the analysis reveals that the LID state fusion method also contributes to a reduction in error counts for monolingual Mandarin and monolingual English utterances, although to a lesser extent compared to code-switching utterances. This observation suggests that the incorporation of language identity information helps the ASR system better capture language-specific patterns and transitions, leading to improved recognition accuracy even in monolingual contexts.

We further analyze code-switching transcription examples generated by the two ASR models: the Transformer-based ASR model (referred to as the baseline model) and the Transformer-based ASR model employed the LID state fusion method (referred to as the fusion model). The results are presented in Table 3. Overall, the transcriptions generated by the fusion model demonstrated superior semantic and grammatical accuracy compared to those produced by the baseline model, particularly when the monolingual segment context was short. These findings underscore the effectiveness of the LID state fusion method in addressing the challenges associated with code-switching speech recognition. By leveraging the contextual cues provided by language identity information, the ASR system becomes more proficient at distinguishing between languages and producing accurate transcriptions. The successful transcriptions of code-switching utterances highlight the LID state fusion method's ability to mitigate language confusion and enhance the system's performance in complex linguistic environments.

## 7 Conclusion

In this paper, we presented a novel language identity-language model scheme to predict language identity from pure text data, eliminating the need for reliance on speech data. We also explored two innovative methods to effectively incorporate text-derived language identity cues into ASR models. Our code-switching speech recognition experimental evaluations on the SEAME corpus demonstrated the effectiveness of our methods. By incorporating language identity information, our ASR system exhibits significantly reduced language confusion in transcribing code-switching utterances, yielding more precise transcriptions for both monolingual and code-switched utterances. Future work includes further enhancements to the LID-LM ar-

chitecture and investigating additional integration strategies to better leverage language identity information in ASR systems. Additionally, exploring pre-training and fine-tuning techniques for LID-LM would open up more possibilities for efficiently utilizing text data to predict language identity.

## Limitations

While our research has made significant strides in advancing code-switching ASR systems through the innovative use of text-derived language identity, there are several limitations that warrant consideration. First, our approach relies on the assumption that text data sufficiently captures language identity information. While this assumption holds for many contexts, it may not fully encompass the richness of linguistic diversity present in certain code-switching environments. Second, the performance of our method may vary across different language pairs, dialects, or domains. The robustness of the approach to such variations would benefit from additional scrutiny and validation in diverse linguistic settings. Lastly, our work explores specific integration strategies (LID state fusion and language posterior biasing) for incorporating text-derived language identity cues into ASR models. There may be alternative strategies or hybrid approaches that warrant exploration in future research to further enhance code-switching ASR systems. These limitations, while acknowledged, should be viewed as opportunities for future investigations to expand upon the foundations we have laid in this study.

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
