# OpenReview forum: "Text-Derived Language Identity Incorporation for End-to-End Code-Switching Speech Recognition"
_EMNLP/2023/Workshop/CALCS — EMNLP 2023 Workshop CALCS_

### Official Review · Reviewer_4pVE · 2023-10-02
**Text-Derived Language Identity Incorporation for End-to-End Code-Switching Speech Recognition**

**Rating:** 2
**Confidence:** 4

**Review:**

Strengths:

This work studies integrating text-derive language identities (LIDs) into the end-to-end ASR system. The authors propose the language identity-language model which incorporates language identity information into the model by inserting a LID token into the beginning of each text token. Then they introduce LID state fusion mechanism to fuse LID and text representations and use a language posterior biasing method to further improve the performance of ASR systems. Experiments and analysis show that the proposed method is effective on English and Chinese languages.

Weaknesses:

1. The proposed language identity-language model is computationally expensive. By inserting LID tokens, the length of model inputs is doubled. The LID tokens are not meaningfully rich and, from the experimental results, it doesn't seem worthwhile.
2. Experiment results show that the improvement from the proposed method is very marginal. Compared to the baseline, most improvement comes from using a transformer model instead of the proposed method. The proposed method only further improves the performance by 1.3% MER at most. Additionally, the compared previous works are not strong. The authors should make a comparison with the most recent works.
3. From the analysis, it looks like the error predictions are mainly about language knowledge such as grammars and synthetics instead of language identities. It is not clear if the proposed method can help improve model predictions.

**Candidate For Best Paper:**

No

**Reason For Best Paper:**

n/a

**Related:**

5: It is very related to the workshop.

---

### Official Review · Reviewer_eSEB · 2023-10-03
**Paper presents the use of LID aids in providing more accurate transcriptions for ASR systems, however the annotated data is not evaluated by annotation metrics, which questions the credibility of dataset.**

**Rating:** 3
**Confidence:** 5

**Review:**

This paper describes a novel approach to predicting language identity from text data and incorporating this information into Automatic Speech Recognition (ASR) systems.

The paper claims that the approach of LID is effective in reducing language confusion when transcribing code-switching utterances. This suggests that their method helps ASR systems provide more accurate transcriptions for both monolingual and code-switched speech.

Pros:
It is an established fact that, adding Language information provides further information about the input text. LID has been widely studied in POS tagging, NER, sentiment analysis and other text/token classification tasks. Similarly, it improved the quality of ASR system too, according to the given paper.

Cons:
In Line 413, the authors have pointed out that, they "manually augmented the transcriptions of the above sets by inserting the corresponding language identity token at the beginning of each text token ". It's essential in research to provide transparency and rigor by reporting annotation metrics such as inter-annotator agreement (e.g., Cohen's Kappa) to assess the reliability of manual annotations. Without these metrics, it's challenging to gauge the consistency and quality of the annotations.

Manual annotation is prone to human subjectivity and potential biases. Annotators may have different interpretations of language switching or language identity, leading to inconsistencies in the dataset. Thus inter-annotator aggrement and other scores are needed to access the quality of the dataset.

**Candidate For Best Paper:**

No

**Reason For Best Paper:**

N/A

**Related:**

5: It is very related to the workshop.

---

### Official Review · Reviewer_5Ssq · 2023-10-04
**The author(s) have introduced a novel approach for language identification using text data and have employed two methods to integrate language identity information derived from this text data into the ASR system.**

**Rating:** 4
**Confidence:** 3

**Review:**

Quality:  4

Clarity: 3 (could be improved)

Originality:  4

Significance: 4


Pros: The author(s) have introduced a novel approach for language identification using text data and have employed two methods to integrate language identity information derived from this text data into the ASR system. They claimed improved ASR system performance based on evaluations conducted on code-switched data involving Mandarin Chinese and English (LID-LM with LID state fusion). The authors pointed out that previous research predominantly relied on speech data and seldom utilized text data to identify language switching. The authors demonstrated that their proposed model, LID-LM, is more effective in language identification compared to previous methods that rely on acoustic features for this task. This finding represents a notable positive aspect of their work.

Cons: The author(s) employed a single dataset that includes code-switched data representing Mandarin Chinese and English. Conducting evaluations on additional corpora can provide further evidence of the effectiveness of the proposed approaches. Additionally, conducting experiments with other code-switched language pairs, such as the Spanish-English Code-Switching Corpus (SECS), would be beneficial for gaining a better understanding of how well the proposed methods perform.

**Candidate For Best Paper:**

No

**Reason For Best Paper:**

N/A

**Related:**

5: It is very related to the workshop.